# Using Quantitative Hormone Monitoring to Identify the Postpartum Return of Fertility

**DOI:** 10.3390/medicina59112008

**Published:** 2023-11-15

**Authors:** Thomas P. Bouchard, Katherine Schweinsberg, Amanda Smith, Mary Schneider

**Affiliations:** 1Department of Family Medicine, University of Calgary, Calgary, AB T3H 0N9, Canada; 2Institute for Natural Family Planning, College of Nursing, Marquette University, Milwaukee, WI 53201-1881, USAmary.schneider@marquette.edu (M.S.)

**Keywords:** estrone-3-glucuronide, luteinizing hormone, pregnanediol glucuronide, urine hormone tests, ovulation, postpartum, breastfeeding

## Abstract

*Background and Objectives*: The Marquette Method (MM) has been used for many years to track the postpartum return of fertility using the ClearBlue Fertility Monitor (CBFM). A new quantitative urine hormone monitor (the Mira Analyzer) was compared to the CBFM in one previous study, and using this pilot data, several women have started to use the Mira Analyzer in the postpartum transition to fertility. *Materials and Methods*: This study was a retrospective, observational case series that analyzed hormone data on the Mira Analyzer during the postpartum period. Participants were invited to share their postpartum cycle and hormone observations. Quantitative hormones in the urine included estrone-3-glucuronide (E3G), luteinizing hormone (LH), and pregnanediol glucuronide (PDG). Data were collected using an electronic survey and an online portal for hormone data. Data collected included participant demographics, menstrual cycle characteristics, and reproductive health history. Hormone range values were calculated, and thresholds were identified that would best predict the first ovulation that led to the first postpartum menstrual period, as well as in transition cycles. Hormone patterns were identified in the context of previous studies. *Results*: Twenty participants contributed data for the analysis. Triggering ovulation before the first period postpartum (Cycle 0) usually required higher LH thresholds than for regularly cycling women. Three different patterns were observed in the return of fertility postpartum: minimal ovarian activity, follicular activity without ovulation, and the early return of fertility. Abstinence rates for avoiding pregnancy with experimental thresholds were calculated. *Conclusions*: Higher LH thresholds in Cycle 0 suggest a decreased responsiveness of the ovaries to LH stimulation from the pituitary. This study replicates postpartum hormone patterns from a previous study. Larger studies are planned to evaluate the effectiveness for avoiding pregnancy using the Mira Analyzer in the postpartum return of fertility.

## 1. Introduction

The return of fertility postpartum is usually delayed by lactation. There is a variable time of amenorrhea followed by follicular activity, a potential first ovulation, followed by a first menstrual period. After the first menses, there are usually several cycles with delayed ovulation and longer cycle lengths. Tracking the natural return of fertility postpartum can be challenging, especially because traditional signs of fertility, like cervical mucus secretions, may not always reflect underlying hormone changes [1]. Moreover, there are at least three different hormone patterns in the months before the first menstruation (Cycle 0) [1], and identifying these patterns is not always straightforward [2,3]. The Marquette Method (MM) has been used for many years to track the postpartum return of fertility [4,5,6,7]. Using the ClearBlue Fertility Monitor (CBFM), the MM postpartum protocol has a typical use effectiveness rate of 95% [4]. However, there are several challenges with the use of the CBFM in the MM postpartum protocol, specifically the need to reset the monitor every 10 days, the limitation of testing only when the CBFM asks (usually starting day 6), a lack of quantitative values to show when estrogen in the urine (E3G) rises and falls, an inability to see lower threshold luteinizing hormone (LH) values, and a lack of the ability to confirm ovulation with progesterone (PDG). These disadvantages may be helped by using a new quantitative monitor, the Mira Analyzer, that was recently compared to the CBFM in a pilot study [8].

Using a quantitative monitor like Mira (Figure 1) could also help in identifying different patterns in the return of fertility postpartum. A study of older quantitative data using the Brown Ovarian Monitor [1] was able to establish three distinct patterns in the return of fertility postpartum: the first pattern involves very little hormone activity for a long period of time before the first ovulation (Figure 2A), the second pattern involves follicular activity without ovulation (Figure 3A), and the third involves a rapid return of fertility (Figure 4A). The purpose of the present study was to identify these three patterns in the return of fertility postpartum using the Mira Analyzer, and to evaluate different thresholds of hormones that predict ovulation.

Two test protocols for avoiding pregnancy were used by participants based on the pilot study [8]. The original test protocol was to avoid intercourse on High and Peak days on the CBFM before the return of the first period (Cycle 0). For the Mira Analyzer, the protocol was to avoid intercourse on days when E3G > 150 today or yesterday and when LH > 11 plus 4 days or 3 consecutive days of LH < 11. A slight revision of the Mira Analyzer protocol in this study used by some participants had a lower E3G level (>100). This was not, however, an interventional study for avoiding pregnancy.

To evaluate different protocol thresholds, this study looked at hormone patterns from postpartum breastfeeding women who used both the CBFM and Mira Analyzer. Quantitative results from the Mira Analyzer helped to elucidate endocrine changes in the postpartum transition described in previous studies [9]. For example, the lower sensitivity of the ovaries to pituitary stimulation [10,11,12,13] may be represented by higher LH levels during lactational amenorrhea (Cycle 0) when compared to regular cycles. As described in detail by McNeilly [9], suckling dampens estrogen’s positive feedback on the Gonadotropin-Releasing Hormone (GnRH)-LH pulse generator, thereby disrupting the regular pulse intervals of 60–90 min. The pulse generator drives follicular secretion.

In two related studies [14,15], Velasquez and colleagues showed that there are larger follicles during postpartum amenorrhea, developed by less potent follicle-stimulating hormone (FSH) isoforms, which, despite their large size, produced minimal amounts of estrogen. With less potent FSH isoforms, lower estrogen levels, and dampened feedback on the GnRH-LH pulse generator, ovulation is suppressed, and the LH surge is more erratic. The erratic activity affects the LH levels and does not lead to ovulation.

The findings of these studies may now be assessed in real time using quantitative hormone monitoring during the postpartum breastfeeding transition. With these new data, we will be able to replicate previous findings regarding the duration of postpartum amenorrhea [16] and the return of ovulation postpartum [17].

## 2. Materials and Methods

### 2.1. Design

These cases were part of a broader retrospective, observational case series on the use of the Mira Analyzer in various circumstances (regular cycles, perimenopause, postpartum) that was approved by the Ethics Board at Marquette University (HR 4276, 4 April 2023). Participants provided consent to share their menstrual-charting data, including their quantitative hormone data from the Mira Analyzer, and were invited to an online Mira portal where health care providers can look at data with their clients online.

Marquette Method teachers (who are health care providers) provided couples with advice on how to navigate the postpartum transition using the Mira Analyzer hormone levels, based on the thresholds suggested in the pilot study [8].

Inclusion criteria were as follows: (1) using Mira and/or CBFM during breastfeeding postpartum, (2) between the ages of 18 and 52 years, (3) English-speaking, (4) not on hormonal contraceptives. Because this was not an effectiveness study for avoiding pregnancy, participants were told that the hormone thresholds used for avoiding pregnancy were experimental.

### 2.2. Data Collection

Demographic and cycle characteristics were entered securely online using Microsoft Forms. Mira hormone data and cycle dates (start and finish) were recorded in the Mira App, and other charting data were uploaded online by the participants.

### 2.3. Statistical Analyses

Demographic characteristics were identified with means and standard deviations. Means and ranges of hormones in Cycle 0 and the first 6 cycles postpartum (when available) were calculated. Hormone values related to the first ovulation postpartum were based on an LH surge identified within a few weeks of the first menses postpartum. Cycles 1–6 postpartum were analyzed based on an estimated day of ovulation (EDO), defined as the last high day of the LH prior to a decrease in the LH of 50% or more. All statistical analyses were conducted using IBM Statistical Package for Social Science software (SPSS Version 28.0, Chicago, IL, USA) and Excel (Microsoft Version 16.78, Redmond, WA, USA).

## 3. Results

### 3.1. Demographic Characteristics

The 20 participants had an average age of 32.5 years, an average of 2.7 children, and an average BMI of 26.3 (Table 1). All were university-educated. The women were followed for 3–12 months postpartum. A total of 15 were totally breastfeeding and 5 were partially breastfeeding (self-report at time of consent).

### 3.2. Postpartum Hormone Patterns

Table 2, Table 3 and Table 4 show the cycle parameters and E3G and LH results of the days leading up to ovulation in Cycle 0 (before the first menses) and transition Cycles 1–6 (there was not enough PDG data to represent in the tables). The sample size for each cycle was not adequately powered to statistically compare the differences in the parameters between cycles; seven participants were still in Cycle 0 at the time of manuscript preparation.

The postpartum transition cycles of three participants are included in Figure 2, Figure 3 and Figure 4 as examples of the three different patterns in the return of fertility postpartum. All three patterns found in the older postpartum study [3] were reproduced in the Mira results. The proportions of women in each group were as follows: (1) ovarian quiescence (Figure 2)—42% of women in the 2018 study and 62.5% in this study; (2) follicular activity (Figure 3)—15% of women in the 2018 study and 37% in this study; and (3) early return of fertility (Figure 4)—42% of women in the 2018 study and 12.5% in this study. Ovulation did not always follow LH thresholds > 11, as found in the pilot with women in regular cycles [8]. In the first two patterns (Figure 2 and Figure 3), there were several LH rises in the 11–20 range that did not lead to ovulation and menses.

Total breastfeeding women were proportionally more likely to be in the ovarian quiescence group (62%) compared to the follicular activity (31%) or early return of fertility (8%) groups. Partial breastfeeding women were proportionally equally likely to be in ovarian quiescence (40%) and follicular activity (40%) compared to the early return group (20%).

### 3.3. Abstinence Rates Using the Marquette Method Postpartum Protocol for Avoiding Pregnancy in Cycle 0

In the original pilot study [8], an E3G threshold of 150 was recommended for advising abstinence in Cycle 0; however, a more conservative threshold of 100 was later recommended based on the E3G levels noted in the 5 days leading up to ovulation. Case 2 (follicular activity with delayed ovulation) highlights the situation of high E3G that gives women many High days on the CBFM, which leads to long periods of abstinence (because the CBFM automatically reads High until a reset is triggered or a Peak reading is determined, even if the E3G levels decrease in the subsequent days).

Abstinence rates were compared using the CBFM Marquette Method postpartum protocol to the difference thresholds of E3G and LH on the Mira monitor. Table 5 shows abstinence rates for various thresholds. The ClearBlue Fertility Monitor protocol had a higher percentage of abstinence days. The Mira protocol with an E3G of 150 and LH of 11 (originally advised protocol) had the lower abstinence rates, but when the fertile-window E3G levels were found to be <150 (see Table 3 in days −5–0 leading up to ovulation), a more conservative threshold of <100 was recommended. Moreover, when the LH thresholds in Cycle 0 were found to be higher than those in regular cycles, a more relaxed LH threshold of 15 was suggested. The combination of E3G 100 and LH 15 had an acceptable percentage of abstinence days and would likely be an improvement on the current Marquette Method postpartum protocol with ClearBlue.

For those who had follicular activity (transient rises in E3G prior to ovulation), it is no surprise that these participants had more abstinence than those with ovarian quiescence, as the latter have fewer high-E3G days.

## 4. Discussion

The three patterns of the return to fertility postpartum that were evaluated based on data from the early 1990s with the original Brown Ovarian Monitor [3] have now been reproduced in a postpartum population of women using the new Mira Analyzer. These three patterns can help women understand what pattern they may follow in their return of fertility while breastfeeding. There is a higher likelihood of follicular activity and the early return of fertility in women who are not totally breastfeeding. The ovarian quiescence pattern (Figure 2) was more likely in women who were totally breastfeeding than in women who were partially breastfeeding, whereas the follicular activity (Figure 3) and early return of fertility (Figure 4) were more likely in women who were partially breastfeeding. Non-lactating women are known to have an earlier return to fertility postpartum [18]. In this small case series, we show that partially breastfeeding also favours an earlier return to fertility as well.

Physiologically, suckling reduces estrogen’s feedback on the pituitary (as opposed to supplemental feeding or expressed breastmilk by bottle) [9]. The reduced feedback on the GnRH from the pituitary dampens the effect on the ovary to secrete follicles. This diminished follicle development also means that the LH threshold to stimulate ovulation needs to be higher. This is precisely what we found in this small case series—the LH thresholds (>11 mIU/mL) that were found to trigger ovulation in regularly cycling women in the pilot study [8] were not adequate to trigger ovulation in breastfeeding women (the LH ranges in Cycle 0 were from 18.5 mIU/mL to 66.1 mIU/mL). This is the first evidence of the higher LH threshold required to trigger ovulation in Cycle 0 before the first menses.

Estrogen is an important indicator leading up to the rise in LH [19], but, in the postpartum period, an estrogen rise can occur without an LH rise, which we have described as “follicular activity with delayed ovulation”, as in pattern 2 shown in Figure 3. This follicular development is likely related to the disconnected feedback loop between the ovary/follicles and the pituitary. Fluctuating estrogen rises have been shown in normal ovulatory cycles [20], and it is possible that there are multiple waves in estrogen excretion from follicles [21]. This wave pattern could also be present in the return of fertility postpartum as well and could eventually be shown in a larger sample size tracking hormones with the Mira monitor. The presence of follicular E3G waves without an LH rise is important to recognize, and to plan for potential fertility with both an E3G rule and an LH rule.

Although an initial threshold of 150 and the revised threshold of 100 for E3G would provide conservative approaches to avoid pregnancy postpartum, it still may involve a lot of abstinence for women with follicular activity without ovulation (Pattern 2, Figure 3). Analysis of the abstinence rates required with respect to different hormone thresholds will have to be weighed against how well these thresholds provide adequate warning of an impending first ovulation. Updated thresholds for a Cycle 0 protocol with the Mira Analyzer are shown in Table 6. A threshold that requires too much abstinence can result in a couple feeling frustrated and may lead to non-compliance, especially if it does not predict the fertile window adequately.

With the above protocol, the abstinence rates would be 36% (see Table 5), which is lower than the abstinence rates in regular cycles using the Marquette Method (40–50% abstinence rates, unpublished data). There would be more abstinence rates when there is more follicular activity (Figure 3, with more estrogen elevation) at 51%, compared to the scenario of ovarian quiescence (Figure 2) at 28%, but the latter is the more common group in this small sample. It should be noted that most participants did not test every single day, and the cost of testing makes this a less likely scenario than intermittent testing. In the context of intermittent testing, a 2-day consecutive span of testing with E3G < 100 and LH < 15 is likely still reasonable but will require a larger sample size to establish survival rates for avoiding pregnancy. Another important assessment tool in future survival trials would be ease-of-use and acceptability measures, as we have used in previous studies in the postpartum period [5].

Another option for quantitative homone evaluation in Cycle 0 is the use of follicle-stimulating hormone (FSH) in these early postpartum cases. The use of FSH could provide greater insight into the underlying follicular development of the postpartum breastfeeding woman in general, as well as into distinguishing the three different return-of-fertility patterns previously described. The FSH test stick cost may pose a challenge, even if benefits are observed. Initial testing should be considered to determine whether there is a benefit for predicting the return of fertility, weighing this against the amount of testing and associated costs that would be required, and the overall implications for postpartum fertility management.

In the present study, urinary pregnanediol-3-glucuronide (PDG) was measured by some women, but there were not sufficient tests to identify an appropriate threshold for the confirmation of ovulation in this small group of women. It is a promising approach, however, as progesterone has previously been used to identify the first ovulation postpartum [22]. Future studies should incorporate PDG tests after LH rises to confirm whether ovulation has occurred.

Although prolactin is a relevant hormone for lactation early in the postpartum transition, there is variability in the amounts of prolactin with suckling, as well as when ovulation resumes [9], so it is likely that different women have different sensitivities to the prolactin concentration [23]. There is currently no urinary prolactin measure, but, in the future, this may be an additional tool to track the return of fertility in lactating women.

Many of the challenges with the CBFM-based MM postpartum protocol have been overcome with the new Mira Analyzer, which does not require 10-day reset cycles, allows for testing on any day, and allows for same-day retesting. Moreover, the levels of E3G and LH can be interpreted in context, and the PDG levels confirm when ovulation has happened. Abstinence rates with the Mira Analyzer were less than those for the CBFM (Table 5), and this is likely because transient E3G rises can be more easily tracked with Mira. These transient rises in E3G are expected to be less in the early postpartum period compared to after 6 months when supplementary solid foods are added and the return of ovulation is imminent, although this was not specifically analyzed in this dataset.

It is important not to over-generalize the results from this small study, understanding that this is only a pilot study in preparation for larger effectiveness studies using quantitative monitors in the postpartum return of fertility. Other important considerations in larger studies include the impact of the BMI and nutritional status, which were shown to play a role in the return to fertility in lactating women in one study [24]; however, in another study, the BMI did not seem to be a factor [25]. In this study, one participant had a BMI > 40 and another three had BMIs between 35 and 40, but all four were still in Cycle 0. In the future, larger studies could evaluate whether the BMI has an impact on the parameters of the return of fertility. Moreover, specific suckling behaviours [26] and other differences exist between different countries [27,28], so the application of the results in this North American population may have to be interpreted differently in different contexts. In the future, breastfeeding patterns (e.g., partial vs. total, duration) could be analyzed in relation to the three patterns of the return of fertility.

## 5. Conclusions

In Cycle 0, higher thresholds of LH are required to trigger ovulation, so a higher threshold of LH (i.e., LH < 15), along with a more conservative E3G threshold (i.e., E3G < 100) to reflect the average E3G levels within the fertile window of Cycles 0–6 postpartum (Table 3), could be used in Cycle 0 for avoiding pregnancy (Table 6). This protocol will be the basis of a larger effectiveness study using the Mira Analyzer for avoiding pregnancy postpartum.

## Figures and Tables

**Figure 1 medicina-59-02008-f001:**
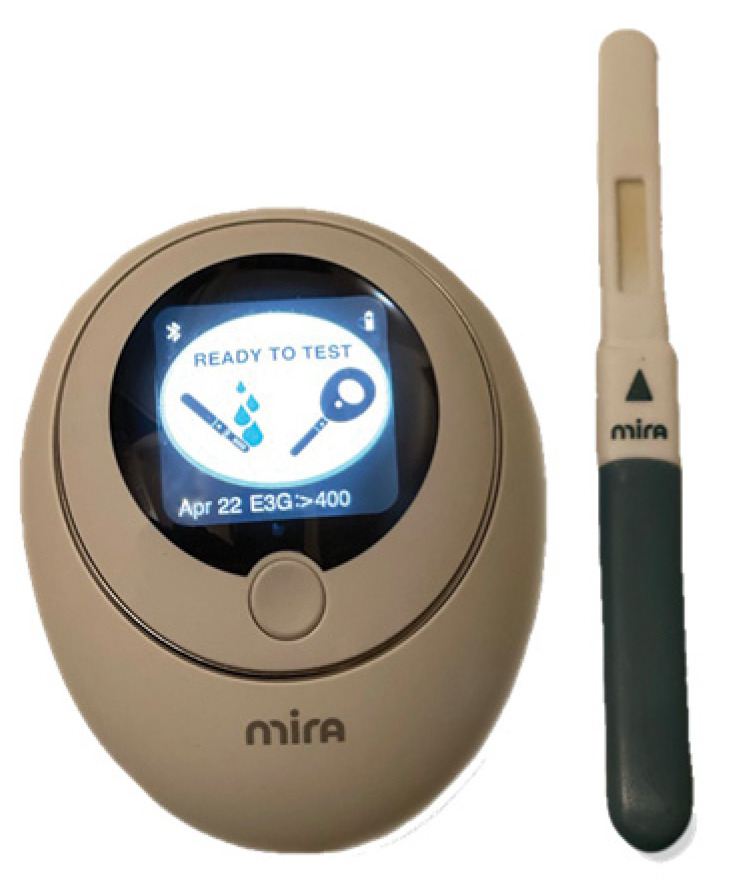
The Mira Analyzer measures follicle-stimulating hormone (FSH), estrone-3-glucuronide (E_1_3G), luteinizing hormone (LH), and pregnanediol glucuronide (PDG) in the urine. It is comparable in identifying hormone changes to the well-known ClearBlue Fertility Monitor. This figure is a partial reproduction of Figure 1 from ref. [3].

**Figure 2 medicina-59-02008-f002:**
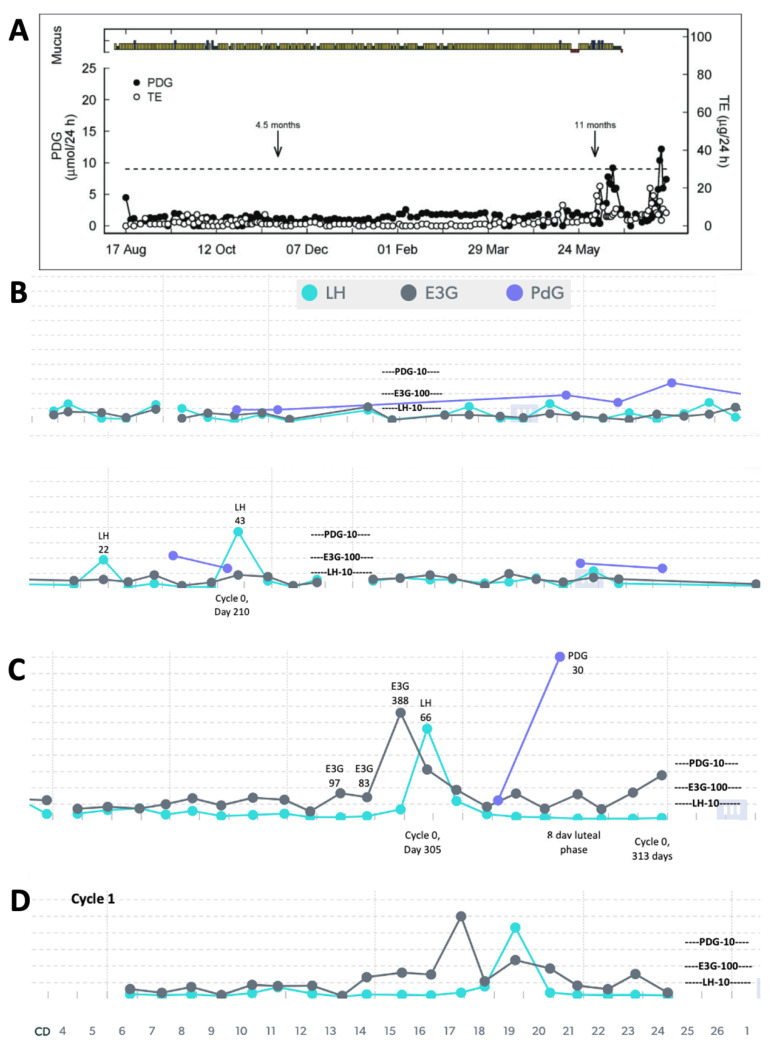
(**A**) Reproduction of Figure 2 from Bouchard et al. 2018 [1] showing ovarian quiescence with delayed ovulation (mucus/bleeding colours—green: dry; yellow: non-lubricative sticky mucus; blue: lubricative, clear, stretchy mucus; red: bleeding/menstruation; TE: total urinary estrogens; end of full breastfeeding with introduction of solids indicated by arrow at 4.5 months, and complete weaning indicated by arrow at 11 months). Mira Analyzer data in (**B**,**C**) show very low estrogenic activity with LH rises (marked levels) that did not lead to ovulation. In (**C**), a high LH surge of 66 was required to trigger ovulation, confirmed with a PDG rise (marked) and the first menses, leading to the first menstrual cycle in (**D**).

**Figure 3 medicina-59-02008-f003:**
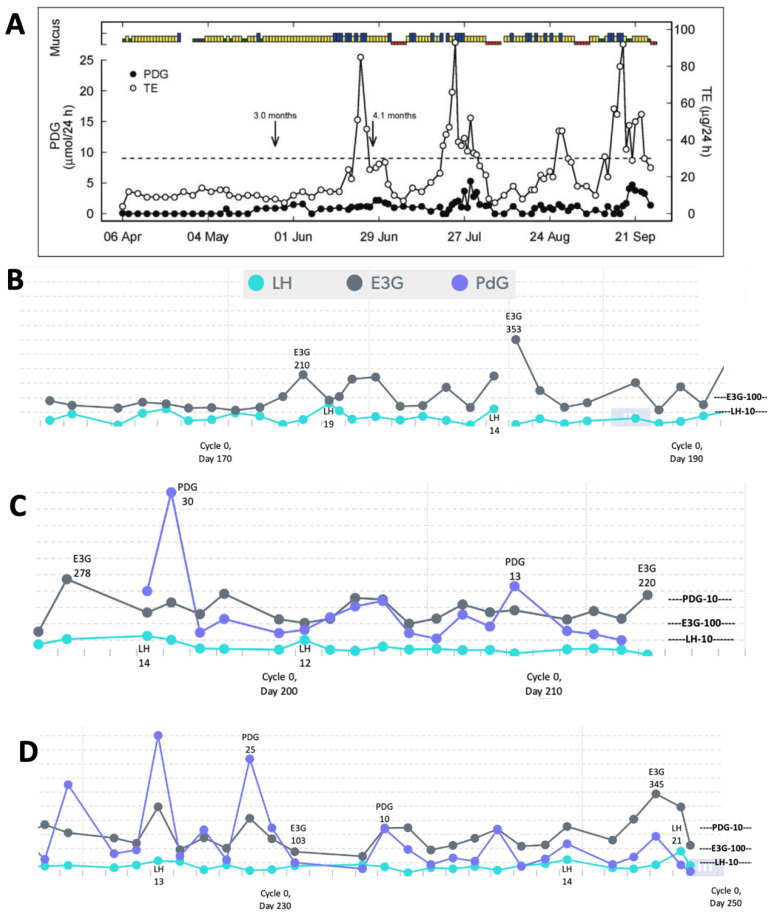
(**A**) Reproduction of Figure 3 [1] showing follicular activity with delayed ovulation (mucus/bleeding colours—green: dry; yellow: non-lubricative sticky mucus; blue: lubricative, clear, stretchy mucus; red: bleeding/menstruation; TE: total urinary estrogens; end of full breastfeeding with introduction of solids indicated by arrow at 3.0 months, and complete weaning indicated by arrow at 4.1 months). (**B**–**D**) Cycle 0 data from the Mira Analyzer show multiple waves of high estrogenic activity throughout, and some LH levels that are relatively high but did not lead to ovulation. Progesterone rises are erratic and not correlated with ovulation with no menses yet.

**Figure 4 medicina-59-02008-f004:**
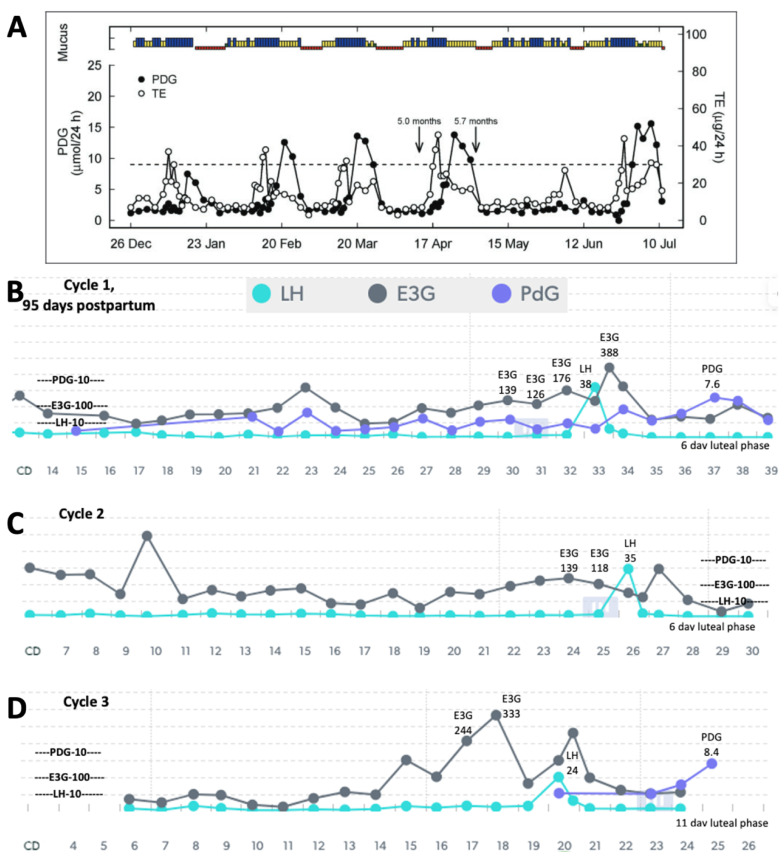
(**A**) Reproduction of Figure 4 [1] showing early return of fertility postpartum (mucus/bleeding colours—green: dry; yellow: non-lubricative sticky mucus; blue: lubricative, clear, stretchy mucus; red: bleeding/menstruation; TE: total urinary estrogens; end of full breastfeeding with introduction of solids indicated by arrow at 5.0 months, and complete weaning indicated by arrow at 5.7 months). (**B**) The first cycle postpartum with Mira, showing high estrogen in the fertile window leading up to a high LH surge and a weak progesterone response with a short (6-day) luteal phase. Cycle 2 (**C**) shows an appropriate estrogen rise in the fertile window, an LH surge, and a 6-day luteal phase. Cycle 3 (**D**) shows an improved follicular estrogen rise, a normal LH surge, and a higher PDG response with an 11-day luteal phase (i.e., gradual return to normal fertility). The longer luteal phase in the third cycle suggests improved chances of fertility.

**Table 1 medicina-59-02008-t001:** Demographic characteristics.

	Mean	SD	Range
Age	32.5	5.4	22–43
BMI	26.2	7.1	19.6–45.6
Pregnancies	3.2	2.0	1–8
Miscarriages	0.3	1.0	0–4
Children	2.7	1.4	1–6
	**Description**		
Education	2—Some College; 9—Bachelor’s; 6—Graduate		
Racial/Ethnic Background	16 (80%)—White; 2 (10%)—Black; 2 (10%)—Latina		
Breastfeeding Status	15 (75%) Total; 5 (25%) Partial		

**Table 2 medicina-59-02008-t002:** Menstrual cycle lengths and peak days. Eight participants were still in Cycle 0 and so did not contribute to the statistics in this table. Two participants did not have Cycle 0 data.

	Peak Day	Length	Luteal Phase
Cycle 0 (n = 10)	218.6 ± 65.9 (123–320)	223.8 ± 71.6 (125–320)	5.7 ± 2.7 (2–10)
Cycle 1 (n = 10)	25.4 ± 7.4 (12–36)	33.7 ± 7.4 (22–47)	8.2 ± 2.8 (4–12)
Cycle 2 (n = 8)	22.9 ± 5.4 (15–31)	32.5 ± 5.1 (25–42)	9.4 ±2.7 (6–13)
Cycle 3 (n = 8)	25.3 ± 6.0 (19–33)	35.0 ± 6.0 (28–45)	9.8 ± 2.7 (5–13)
Cycle 4 (n = 5)	18.0 ± 2.0 (15–20)	29.6 ± 3.0 (26–34)	11.6 ± 2.7 (7–14)
Cycle 5 (n = 4)	15.5 ± 2.4 (13–18)	26.8 ± 4.6 (22–33)	11.3 ± 2.6 (9–15)
Cycle 6 (n = 2)	17.0 ± 2.8 (15–19)	29.0 ± 2.8 (27–31)	12.0 ± 0 (12)

**Table 3 medicina-59-02008-t003:** Estrone-3-glucuronide (E3G) values in ng/mL for days leading up to estimated day of ovulation (EDO = 0).

	−6	−5	−4	−3	−2	−1	0	1
Cycle 0 (n = 10)	89 ± 58	105 ± 41	117 ± 63	159 ± 101	216 ± 132	255 ± 103	177 ± 176	129 ± 51
Cycle 1 (n = 10)	133 ± 162	142 ± 125	126 ± 73	148 ± 71	161 ± 76	155 ± 71	239 ± 180	236 ± 190
Cycle 2 (n = 8)	95 ± 55	111 ± 80	129 ± 62	119 ± 50	165 ± 60	216 ± 113	251 ± 191	300 ± 242
Cycle 3 (n = 8)	127 ± 76	110 ± 71	157 ± 97	190 ± 64	284 ± 142	207 ± 81	296 ± 189	197 ± 136
Cycle 4 (n = 5)	100 ± 58	88 ± 64	137 ± 33	133 ± 79	181 ± 106	212 ± 108	252 ± 229	167 ± 65
Cycle 5 (n = 4)	190 ± 170	166 ± 172	138 ± 89	135 ± 117	186 ± 88	232 ± 118	367 ± 108	128 ± 84
Cycle 6 (n = 2)	162 ± 73	116 ± 78	227 ± 249	106 ± 71	199 ± 88	294 ± 143	331 ± 83	175 ± 94

**Table 4 medicina-59-02008-t004:** Luteinizing hormone (LH) values in mIU/mL for the days leading up to estimated day of ovulation (EDO = 0).

	−6	−5	−4	−3	−2	−1	0	1
Cycle 0 (n = 10)	3.2 ± 1.5	4.5 ± 1.5	5.4 ± 3.9	4.8 ± 2.5	6.8 ± 6.1	16.1 ± 11	46.2 ± 18	11.0 ± 8.2
Cycle 1 (n = 10)	3.0 ± 2.2	3.2 ± 1.4	3.0 ± 1.2	3.1 ± 1.8	3.6 ± 1.4	7.0 ± 6.0	48.2 ± 26.7	8.7 ± 6.8
Cycle 2 (n = 8)	4.2 ± 2.3	4.7 ± 2.8	4.7 ± 3.5	2.6 ± 1.2	3.7 ± 1.5	8.3 ± 8.2	30.7 ± 16.7	7.9 ± 4.2
Cycle 3 (n = 8)	7.8 ± 9.0	5.3 ± 4.3	4.1 ± 1.6	3.5 ± 1.7	5.0 ± 2.4	9.9 ± 5.7	57.6 ± 49.8	9.7 ± 7.1
Cycle 4 (n = 5)	3.5 ± 0.9	3.4 ± 3.0	3.7 ± 2.0	3.7 ± 1.8	4.1 ± 2.6	4.0 ± 1.4	49.0 ± 17.8	15.5 ± 11
Cycle 5 (n = 4)	4.1 ± 2.8	5.9 ± 4.4	4.4 ± 3.1	3.4 ± 1.9	4.2 ± 2.7	2.0 ± 0.7	42.4 ± 24	8.1 ± 2.6
Cycle 6 (n = 2)	4.1 ± 1.3	3.6 ± 2.4	3.4 ± 3.3	2.8 ± 1.9	2.8 ± 2.3	3.3 ± 3.2	31.6 ± 27.5	14.3 ± 9.5

**Table 5 medicina-59-02008-t005:** Abstinence rates for the CBFM Marquette Method postpartum protocol compared to various Mira E3G (“today or yesterday”) and LH (LH > 11 plus 4 days) thresholds. The Early Return of Fertility category had only one participant with Cycle 0, so it is not represented here. Three participants did not have Cycle 0 data. One participant had an excessive number of false LH surges, so abstinence calculations are not included here.

	CBFM	Mira E3G 150 LH 11	Mira E3G 100 LH 11	Mira E3G 150 LH 15	Mira E3G 100 LH 15
All Cycle 0 (*n =* 15)	61%	23%	41%	16%	36%
Ovarian Quiescence (*n =* 10)	53%	20%	34%	14%	28%
Follicular Activity (*n* = 5)	71%	33%	58%	22%	51%

CBFM: ClearBlue Fertility Monitor; E3G: estrone-3-glucuronide; LH: luteinizing hormone.

**Table 6 medicina-59-02008-t006:** Revised Cycle 0 protocol with updated thresholds based on the current pilot data. This is based on users testing daily.

	Available Day for Intercourse?
If E3G is <100 today and yesterday AND If LH is <15 today and the last 4 days	Yes
If E3G is ≥100 today or yesterday	No
If LH is ≥15 today or the last 4 days	No

## Data Availability

Data are contained within the article.

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
