# Peer review of "Using Quantitative Hormone Monitoring to Identify the Postpartum Return of Fertility"

_medicina, 2023, doi:10.3390/medicina59112008_

Round 1

Reviewer 1 Report

Comments and Suggestions for Authors

28-33 Conclusion repeats results. Reformulate. Maybe altogether. 

34. add estrone -3-glucuronide, pregnanediol glucuronide

39 activity that could arrive to a bleeding or end in an initial ovulation 

82 would help to elucidate

83-88 Under discussion, results do not go in the introduction.

59-62 refer to a figure that shows the 3 patterns, Explain arrows for breasr feeding status, mucus colors (could be green: dry, yellow; non lubricative sticky mucus, blue: lubricative clear stretchy mucus) red: bleeding/menstruation. Put TE is total urinary estrogens in legend. Do not repeat in the legend the comparative percentages already described under results in the text.

figures 2, 3, 4 BCD: if available put arrows to signal breastfeeding status Explain colors of date file in legend. Suggest use of dark blue for Lh peak, light blue for potentially fertile period, brown (avoid green or yellow) for infertile days. red for bleeding/menstruation, not pink. Explain in legend this is retrospectively considered. Would add a prosprective file of potential fertility, could be  over CD file (explicit CD is cicle day in legend) with best threshholds for EG and LH using same colors suggested.

143 or for?

153 table 2: maybe those 8 participants pending could be included after cycle 0 when available. Explicit LH Peak day (mean, SD, range). Add luteal phase SD and range.

136 breastfeeding status is dynamic, seems considered at entrance. Explicit 

Repeated text in the table

155 specify EDO (0)

233  ? breastfeeding status is not in the results

295 would add relevance of ecographic serial study to correlate hormonal data, evaluate disovulation and treshholds described, Specially those of PDG that lack lack this (LUF?).  

Also would consider in results and discussion relationship with LAM (lactational amenorrea) infertility using part of this data set (total breastfeeding, no bleeding, up to 6 moths postpartum). If there are some cases available.  

Author Response

Thank you for your very helpful comments.

1) We have reformulated the Conclusion in the abstract as suggested to avoid duplicating the Results.

2) Urine hormone metabolite names were added.

3) Progression in Cycle 0 was clarified, specifically that there may not be ovulation ("potential ovulation" was used).

4) We elucidated that the test protocol came from the pilot study.

5) This is appropriate for the introduction because it was used by the research participants and results from the test protocol was included in the results (not results but came from original pilot study).

6) Legends of Figures 2-4 have been revised as suggested.

7) For Figures 2-4, date rows have been removed as the colours produced in this row were not relevant and can cause confusion (automatically generated by Mira App and not relevant to our tracking). The suggestion about shaded days for the fertile window is thoughtful but potentially premature given these remain test protocols that have not yet been validated in an effectiveness study.

8) For Table 2, we had SD and range to the luteal phase. We were able to add one more case for Cycle 0 and Cycle 1. LH peak day is in bold in Table 4.

9) Breastfeeding status is now stated explicitly: self-report at time of consent.

10) Table 1 duplication of text was removed to shorten text.

11) EDO=0 was reported in Tables 3 and 4.

12) Breastfeeding status is indicated in section 3.1 and Table 1.

13) A comment in the discussion was added with respect to analyzing breastfeeding patterns in more detail.

Reviewer 2 Report

Comments and Suggestions for Authors

Reviewer 1

 Using Quantitative Hormone Monitoring to Identify the Postpartum Return of Fertility. Thomas P. Bouchard et al.

 Summary comment:

 This study is important for two reasons:

a)    It provides day-specific E3G and LH, and to a more limited extent PDG levels, throughout the postpartum interval. Such published data are virtually nil.

b)    It attempts to solve an important still existing problem in natural family planning, the identification of the return to fertility during lactation.  Preliminary thresholds for E3G and LH which can be used to signal potentially fertile intervals are proposed.

Recommendations and Criticisms:

 1)    One important conclusion of this work is the quantitation of the three general forms of ovarian function in the postpartum/lactation state: ovulatory quiescence, follicular activity with unsustained estrogen levels and without ovulation, and rapid return to normal ovulatory function. I would better define this in the abstract.

2)    Figure legends for 2 – 4 lack clarity and should be improved. I recommend the first sentence of each figure as a descriptive title. For example for Figure 2: ‘E3G, LH, and PDG tracking of a quiescent postpartum cycle (Subject 1). Mira analyzer data are shown for Subject 1 with low estrogenic activity (B) followed by E3G and LH peaks, a rise in PDG, and menses (C).’ I would then put in some other noteworthy points. I understand what you are doing with the Bouchard et al. 2018 graph – showing the three forms of ovarian function – but it is confusing. I would move it to the bottom of each figure and provide more explanation. I would define TE, etc.

 3)    You note that the BMI of some of your subjects was up to 45.6. I wonder if patients with such a BMI should have even been included in this initial preliminary study, given the incidence of polycystic ovary syndrome at high BMI and underlying problems with ovarian function. You do mention this at the very end, but I would put in some more qualification about how this might affect the results.

 4)    The use of E3G thresholds to define the fertile window is still problematic. This plagues precision and effectiveness for NFP, at least in some cases. Preovulatory LH surges and unsustained estrogen rises occur not infrequently during normal ovulatory cycles, not just in the postpartum (this phenomenon referenced in: Usala, S.J.; Alliende, M.E.; Trindade, A.A. Algorithms with Area under the Curve for Daily Urinary Estrone-3-Glucuronide and Pregnanediol-3-Glucuronide to Signal the Transition to the Luteal Phase. Medicina 2022, 58, 119. https://doi.org/10.3390/medicina58010119 and in References 13 – 17 therein.) An explanation for this may be the new model for ovarian function: follicular development occurs in multiple waves even during normal ovarian function (see Baerwald AR, Adams GP, Pierson RA. Ovarian antral folliculogenesis during the human menstrual cycle: a review. Hum Reprod Update. 2012 Jan-Feb;18(1):73-91. doi: 10.1093/humupd/dmr039. Epub 2011 Nov 8. PMID: 22068695). It would probably be worthwhile to mention this.

 5)    There appears to be an error in Table 4, Cycle 2, the column for -1. The LH is reported to be 7.4 ± 42.2.

Author Response

Thank you for these helpful comments.

1) The three patterns of return of fertility were specified in the abstract.

2) The legends for Figures 2-4 were revised to provide more clarity. TE abbreviation is defined, more detail about hormone changes were added. We opted to keep the reproduced figures at the top for easy of interpretation.

3) Thank you for the insightful comment about BMI, the following was added to the discussion: "In this study, one participant had a BMI > 40 and another three had a BMI between 35-40, but all four were still in Cycle 0. In the future larger studies could evaluate whether BMI has an impact on the parameters of return of fertility."

4) Excellent comments. These two papers were added with a comment in the discussion.

5) This SD error was fixed, thank you for identifying this.

Round 2

Reviewer 1 Report

Comments and Suggestions for Authors

I would include in the figures part A the explanation of the arrows that show la breastfeeding status

Author Response

Thank you very much, we have included this in the figure legends as requested.